# Keeping the power on to home medical devices

**Richard Bean** ⓘ *, **Stephen Snow** ⓘ, **Mashhuda Glencross, Stephen Viller, Neil Horrocks**

Centre for Energy Data Innovation, University of Queensland, Brisbane, QLD, Australia

* r.bean1@uq.edu.au

**Data Availability Statement:** The data repository files are available at https://doi.org/10.14264/uql.2020.802.

**Funding:** This research was funded by the University of Queensland, Advance Queensland AQPTP01216-17RD1) (Queensland Government)

## Abstract

Advances in digital health technologies have revolutionised home medical care. Yet many home medical devices (HMEDs, which includes devices referred to as 'life support equipment') rely upon a stable and resilient electricity supply. For users of HMEDs, interruptions to electricity supply can compromise treatment, well-being or survival. This paper addresses a challenge critical to the continued innovation in digital health technologies: the reliable supply of electricity. We bridge the current gap between electricity networks and digital health technologies through a novel method for the remote detection of the phase (that is, which part of the network that each house is connected to), in order to eliminate avoidable interruptions to supply for HMED users. We present an unsupervised phase identification algorithm capable of remote phase detection at scale, and without transformer data. This method translates data insights into actionable energy provision for HMED users and other vulnerable customers, enables more accurate management and planning, and improves electricity reliability which is critical for HMED users and the continued advances in digital health technologies.

## Introduction

Advances in digital health technologies are revolutionising healthcare. The total global market for mHealth was calculated in 2015 to be worth $21.5 billion [1]. A key impact of the proliferation of remote and mHealth care is the number of people being able to access hospital treatment at home. Electricity is fundamental to the basic function and advancement of digital health technologies (DHT), powering home medical devices (e.g. machines for dialysis, physiotherapy, oxygen delivery or CPAP machines), enabling tele-medicine, and charging devices with batteries. Yet discussions of electricity and access to electricity are largely absent from academic discourse on DHTs to date, given the relative ubiquity and stability of supply of electricity in developed countries. However, the consequences of a disruption to supply for those reliant on certain DHTs can be harmful or even fatal [2, 3].

In the design of DHT where batteries are involved, (e.g. wearables, smart phone apps, mobility aids), power consumption may commonly be considered within the context of extending battery life and minimising the need for frequent charging [4]. Yet contingencies or preparedness for the loss of ability to charge, or the function of DHTs in the absence of access

and Redback Technologies. Authors RB, SS, MG, SV, NH all received funding in the form of salaries from the University of Queensland. The funders had no role in study design, data collection and analysis, decision to publish, or preparation of the manuscript.

**Competing interests:** The authors have read the Journal's policy and the authors of the manuscript have the following competing interests. Author RB and the Centre intend to investigate patenting the clustering algorithm mentioned in the manuscript and may potentially benefit financially from it in the future. This does not alter our adherence to PLOS ONE policies on sharing data and materials.

to electricity is rarely considered outside of situations in which DHTs are designed specifically for environments in which data (WIFI/3G) is lacking (e.g. remote locations or developing world contexts [5]).

Most other DHTs rely on continuous power supply. A recent US study identified over 685, 000 *electricity dependent* users who reside at home [2]. Electricity dependence is defined by vulnerable populations ". . . who depend on durable medical equipment (DMEs) that are electrically powered" [2]. This includes those reliant on electricity for independence, (e.g. electric wheel-chairs, mobility scooters etc), as well as those reliant on electricity for survival, (e.g. those with ventilators, oxygen concentrators, reliance on exceptional temperature stability or other critical at-home medical devices). A much larger subset of the population may additionally be considered electricity vulnerable, such as those susceptible to heat/cold, or with limited mobility to leave home in a blackout [6].

Existing meta reviews of digital health technologies focus on communications [7], patient and family factors in implementation [8]and reporting quality [9]. Yet despite being fundamental to the majority of DHTs, stable electricity supply or contingency for loss of supply remains a knowledge gap.

With advances in DHT technology, tele-medicine, in-home medical devices, and support for independent living, enabling more people to remain in-home for longer, the number of electricity dependent and electricity vulnerable people is projected to increase significantly in coming years [2, 3]. Given the scientific consensus that climate change is likely to cause an increase in extreme weather events [10], increased intensity of hurricanes, prevalence of extreme heat and extreme cold events, and increased severity of flooding [11], it can also be extrapolated that weather-related disruptions to electricity supply will become more prevalent in the years ahead. Already, large-scale blackouts and extreme weather events result in significant spikes in emergency hospital admissions from those losing operation of in-home medical devices [12–14]. The advancement of DHT as a research agenda, is therefore dependent on its ability to withstand interruptions to the electricity supply upon which many DHTs rely; underscoring the salience of research which bridges the current research gap between DHT design, energy network management and disaster response.

This paper represents the foundations for these bridges, arguing for a need to factor emergency preparedness into DHT design. We describe the design and testing of a method for minimising the risk of network operators accidentally disconnecting customers with home medical devices. Through the use of a novel data-driven machine learning approach, we detect the phase connection to domestic homes, enabling network operators to verify if a HMED user is connected to a phase that might be interrupted and action a suitable response to mitigate potential harm. We report on the performance testing of this algorithm before discussing (1) the need for closer collaborations between designers of medical devices, energy network operators and health policy makers, and (2) further potential values of detailed energy usage information required for this algorithm for vulnerable energy users and at-home medical device operators.

## Background

Digital Health Technology (DHT) affords a growing ability for medical patients to receive care at home. In 2004 it was estimated that over seven million US residents received home health care annually, many of which involved the use (or continuing use) of in-home medical devices [15, 16]. Yet the guidance available to users for power outage for in-home medical devices is limited. With regard to electricity supply, "Design Considerations for Devices Intended for Home Use", aimed at manufacturers of home-medical devices states only that designers and

manufacturers might ". . . consider providing or identifying backup power options, such as a battery or generator", and instructions on the device labelling for emergency contact information in the event of an outage [16].

## Treating electrical emergencies

While hospitals and other critical infrastructure are typically fitted with onsite backup generation for use during blackouts, multiple studies have demonstrated the substantial effect of power disruptions on people with home medical devices. Rubin and Rogers [17] highlight how hospitals in Louisiana following Hurricane Isaac in 2012 were essentially treating "electricity emergencies" rather than medical emergencies; namely people who could no longer operate electricity dependent medical or enabling devices. Libraries and shelters set up by charities acted as "electricity shelters", reducing the burden on hospitals. People with home medical devices accounted for 22% of all hospital admissions in a 24 hour period following the 2003 New York blackouts [13]. Due to the "en-masse" movement of home medical device patients to local hospitals following the 2011 Japan earthquake, authors conclude such events have the potential to "overwhelm the capacity of hospital inpatient facilities" [12]. Even planned load shedding events in South Africa have been correlated (with causation established) to a 10% increase in hospital admissions, including from medical device patients and those with secondary effects, including carbon monoxide poisoning from petrol generators and food poisoning from lack of refrigeration [18].

In response, authors argue for the need for vulnerability assessments of power grids, understanding the likely nature of blackouts both physically (i.e. which areas will lose power first/last) and socially (i.e. what are the demographics of these areas, where are registered medical device users located) [19].

## Existing preparedness: The Australian network context

In Australia, the retail electricity environment is governed by the National Electricity Retail Rules (NERR) and the National Electricity Retail Law (NERL) [20]. Published in February 2019, the NERL has specific provision for the conduct of energy service providers (e.g. network operators) around *life support equipment*. The Law defines life support equipment as one of the following:

- an oxygen concentrator

- an intermittent peritoneal dialysis machine

- a kidney dialysis machine

- a chronic positive airways pressure respirator

- Crigler-Najjar syndrome phototherapy equipment

- a ventilator for life support

- any other equipment that a registered medical practitioner certifies is required for a person residing at the customer's premises for life support

Users with HMEDs can apply for recognition as a life support customer with their electricity retailer or network operator. Yet it is unlikely that these records provide a complete picture of electricity dependent users. This is due to: (1) it is unlikely that 100% of home medical equipment users will have identified themselves to their retailer or the network, and (2) people

with short term (e.g. two to three day) loans of medical equipment may be unlikely to bother applying, relative to continual users.

The NERR include the requirement for electricity retailers and network operators to not arrange for de-energisation of the premises associated with a registered life support customer unless they provide advanced notification of planned outages. While weather events, natural disasters and their impact on a network reliability cannot currently be predicted, The Australian Energy Regulator (AER) issues substantial infringement notices for failing to notify electricity dependent customers of planned interruptions of supply. A company may be fined $20,000 (as of 2020) if the company fails to provide at least four business days written notice of a planned interruption to electricity supply [21].

A common cause of accidental disconnection or failure to notify customers is inaccurate records regarding the point at which the customer is connected to the network. Residential power supply in Australia consists of three low voltage electrical conductors (called phases) and customers can be connected to any of these phases. Some households are connected to power poles where two separate power supplies, fed from two entirely different sources, may terminate. In these instances, a household could feasibly be connected to any one of six phases.

Depending on the age of a suburb, the initial allocation of houses to phases can predate computer records by many decades, depending on when the house was constructed. The initial record may have been transmitted from paper to computer records, and further between each generation of computer record (magnetic tape, magnetic drive, solid state, and cloud storage). Transmission errors and errors in recording phases occur frequently and the network operator is sometimes unaware of the actual accuracy of their records. Further, whenever an extreme weather event occurs, houses may be reallocated to different phases in the heat of a crisis. Hence while it is in a utility's best interests to have a completely accurate record of the phase allocation of each house attached to a feeder / transformer, in reality this is rarely possible. The issue of inaccurate connection point data is not unique to Australia, with utilities across the world reporting similar problems [22].

To summarise: (1) power outages cause life threatening situations for electricity dependent populations and "electricity emergencies" can stretch the capacity of local hospitals and medical centres. (2) Despite efforts to identify electricity dependent individuals for emergency preparedness, achieving 100% coverage is highly unlikely, due to some users failing to register. (3) Network data on which houses are connected to which phases is imperfect, increasing the chance that even registered users of HMEDs may be adversely impacted by planned outages and natural disasters. (4) These issues are generalisable globally [22].

In order to address this problem, we provide a proof of concept for a phase detection algorithm which holds potential for substantially reducing the impact of power disruptions to HMED users.

## Proof of concept: Remote phase detection

Ordinarily, to manually check the phase of all the houses on a transformer (there are approximately 100 houses connected to each transformer in urban areas), takes a team of technicians several hours. Hence the ability to carry out this work remotely using data analysis techniques is valuable in both time and money, and minimises the chance of disruption of supply to HMED users. As the benefits of big data analytics become more apparent to network operators they will install more and better power sensing technology at customer premises (including smart meters and IoT power quality devices). It is these new sensors that offer the ability to provide real time information on connection point detail and updates on in-field changes.

## Measurement device

Remote phase detection requires behind-the-meter data capture of voltage, current and impedance at regular intervals of at least one minute. Devices capable of this frequency of data capture include: smart meters, which transmit energy use information to the utility at five to 60 minute increments [23]; home energy monitoring sensors; sensors installed in WIFI-enabled rooftop solar inverters and home battery storage; or custom built sensors, which can provide sub-minute granularity. While smart meters are being rolled out across Europe with many countries now at > 80% saturation [24], smart meter saturation is far lower in Australia, and behind-the-meter sensors capable of sub-minute data capture offer opportunities for phase detection (amongst other benefits). The Redback Technologies Ouija Lite (used in this study) is one such low-cost sensor which captures voltage, current, impedance and power factor data at each house it is connected at one minute resolution. It transmits this data to a cloud service through the use of Sigfox devices (or other 4G telecommunication network devices).

The infrastructure can be represented as in Fig 1.

The *Power Analyzer* device is attached to the house. It measures the energy data at two points *Grid* and *Generation* data) and passes it to the Ouija Lite device. This device then transmits data packets, once per minute, via Sigfox, to Microsoft Azure storage in the cloud. Using a Shared Access Key, the cloud storage data can be downloaded and processed using routines in, for example, the R or Python programming languages.

Generally each transformer in a residential area has three active LV (low voltage) phases connected and each house is then connected to one of these three phases. Electricity network operators balance the number of houses connected to each phase.

## Clustering methods

Remote phase detection algorithms are in their infancy and few examples from peer-reviewed literature exist. Pezeshki and Wolfs [25, 26] proposed a supervised clustering method using correlation to identify the phases of houses in an LV network in Perth, Western Australia. They looked at 51 single phase houses and 24 three phase houses. Voltage data from the houses was collected from the smart meters in each house (15 minute periods). This is the minimum resolution at which consumer smart meters record this data.

In order for this correlation based method to work, transformer level voltage data is also required. At the transformer, the average voltage over the same time period was calculated.

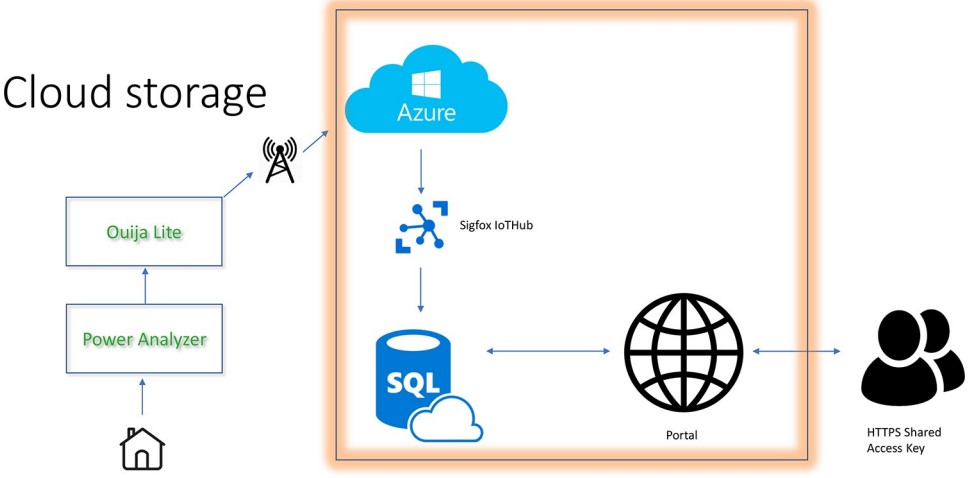

**Fig 1. Cloud infrastructure.**

For each house, the phase with the highest correlation was chosen as the putative phase for that house. This resulted in a correct allocation of phases for each house.

In other work, Wang et al [27] looked at a constrained $k$-means clustering algorithm to analyse a network in Southern California while Wang et al [22] applied a constraint-driven hybrid clustering algorithm, and Blakely et al [28] used spectral clustering.

## Our method

We propose a new algorithm for identifying phases in a group of houses connected to a transformer. The new algorithm is unsupervised and does not require access to transformer level voltage data, as the supervised approach of Pezeshki and Wolf [25, 26] does. Instead, voltage time series data is collected at each of the houses for the purpose of the analysis.

We assess the stability of the clustering with the silhouette values method of Kaufman and Rousseeuw (1987). [29, 30] The technical details of our clustering algorithm are currently in the process of a patent investigation.

We produce a correlation matrix by computing the Pearson correlation coefficient between the time series data available for any two houses. Where a period is missing in any house, this is omitted from the time series vector. From this, a dissimilarity matrix is derived and used as input to the clustering algorithm with $k = 3$ clusters, as there are three phases in the data.

As an illustration of how the correlation coefficient is calculated between houses, we show a voltage time series plot in Fig 2. The houses with voltages represented by the green and red lines are highly correlated, with a Pearson correlation coefficient of 0.81. The house with voltage represented by the blue line is uncorrelated with the other two, with Pearson correlation coefficients of -0.83 and -0.70. In the final clustering, the algorithm places the "green" and "red" houses in one cluster and the "blue" in another.

In a dissimilarity matrix, houses within the same cluster are more similar, while houses in different clusters have their dissimilarity index maximised.

We tested the above method with 68 houses connected to two transformers in a suburban electricity network setting. The exact allocation was provided by the network operator after an audit at each house connection point. Our method correctly allocates 64 of the houses to the putative correct (un-ordered) phase.

We show a diagram of the location of the houses, with blue crosses representing the two tranformers, and red, green and black dots representing the locations of the houses attached to the transformers in Fig 3.

Note that in this paper, we calculate the error rate based on the best ordering of the prediction data, out of the six possible orderings. For instance, in the following confusion matrix, we would assess that $14 + 6 + 9 = 29$ out of 33 houses have been allocated correctly. This is the best of the six possible phase orderings—values from each row and column are chosen so that all rows and columns are used, and the sum of the values is maximized. Reference cluster 1 would be associated with prediction cluster 3, reference cluster 2 with prediction cluster 2, and reference cluster 3 with prediction cluster 1.

|  | *Reference* | | |
| --- | --- | --- | --- |
| *Prediction* | **1** | **2** | **3** |
| **1** | 0 | 2 | 9 |
| **2** | 1 | 6 | 0 |
| **3** | 14 | 0 | 1 |

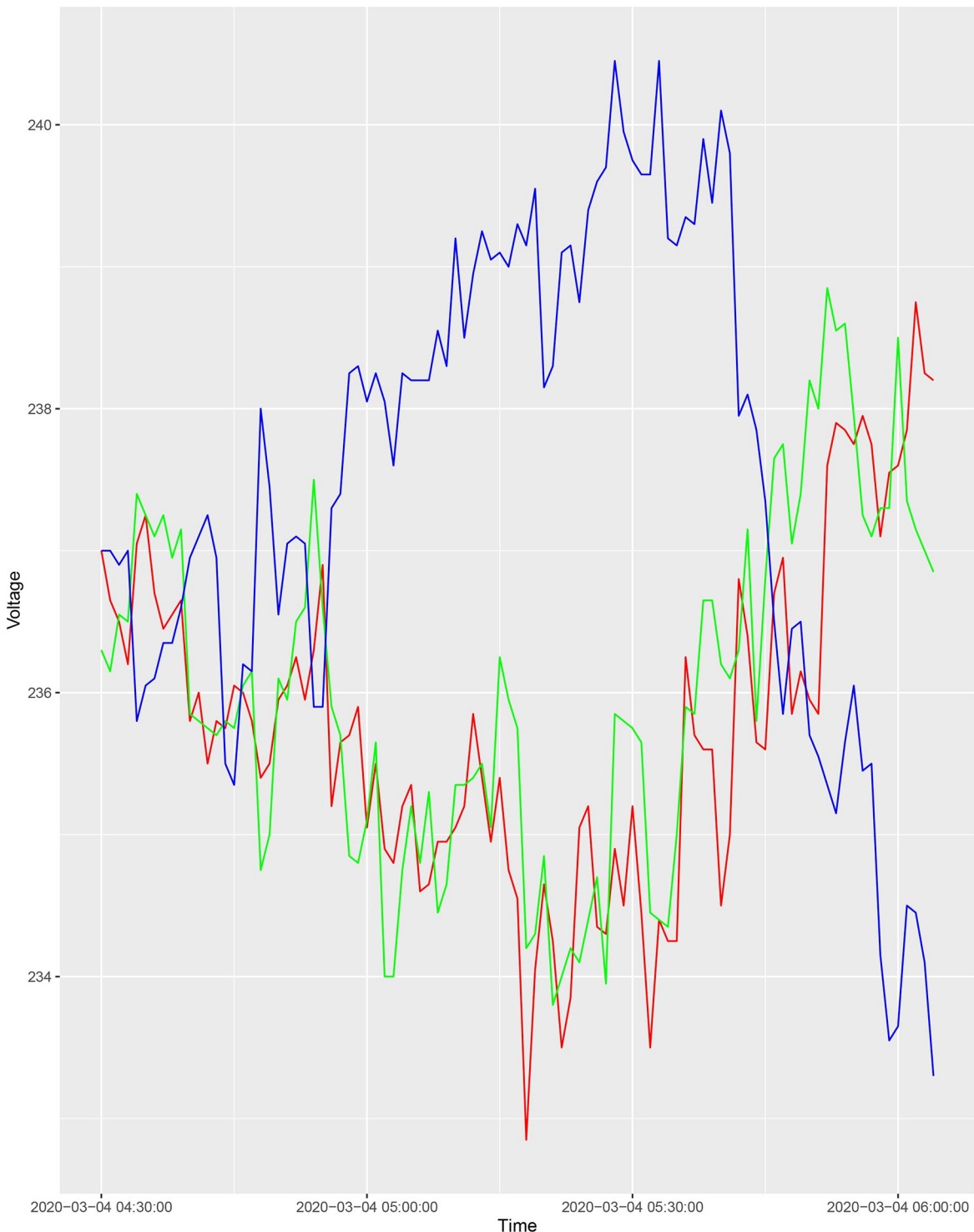

**Fig 2. Voltage time series plot.**

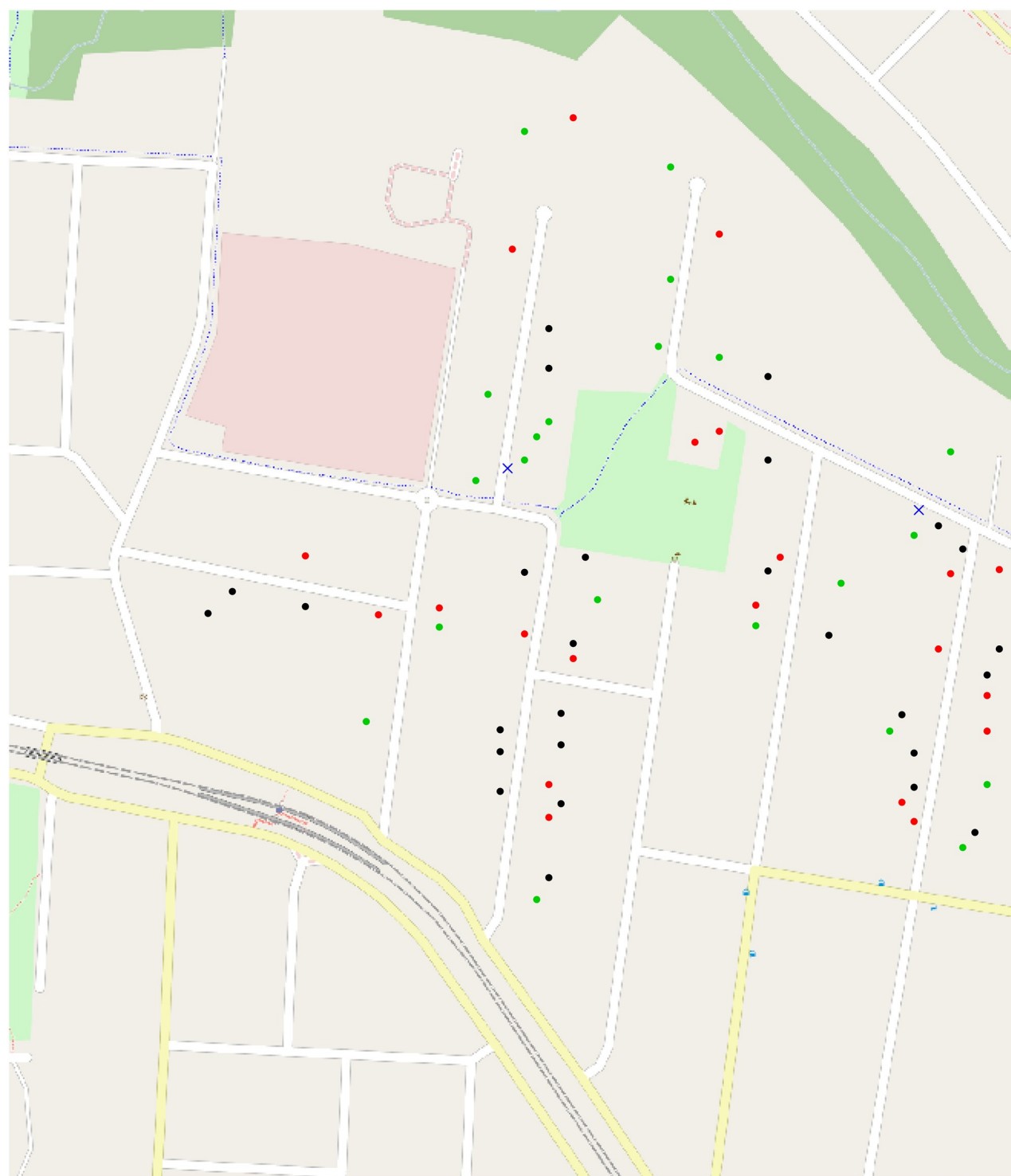

**Fig 3. Locations of transformers and houses in suburb.**

In order to allocate the correct order of phases, with uncertain records we suggest that existing records be leveraged, and testing carried out on a few houses that are clearly uncorrelated and appear to be on different phases.

Transformer 1 has 33 houses attached, while Transformer 2 has 35 houses attached. The data is from 17 January to 23 April 2019 in the first group, and from 5 February to 26 July 2019 in the second group.

At a one minute aggregation level, there are 97,647 rows for Transformer 1 and 179,414 rows for Transformer 2. This corresponds to approximately 70-73% coverage of all minutes in the data.

We plot the correlation matrices for the data and in the following figures (using the *corrplot* package of [31]). We also show the silhouette plots for each group in Figs 4 and 5.

In the first, the clusters produced are of size 11, 7 and 15.

According to the phase allocation provided by the network operator, the houses 12, 29, 41, and 76 are allocated incorrectly. As houses 12 and 29 are firmly within clusters 3 and 2, these may well be record keeping errors. House 29 has other issues as the associated device is allocated to two adjacent addresses in the data. Houses 41 and 76 have lower silhouette values in the chart indicating lower confidence in the clustering.

Fig 6 shows three clear clusters (dark blue squares) representing the three power phases.

To validate the accuracy of our algorithm, the predicted phases for each house was checked against the network operator's records for which house was connected to which phase.

In the second, the clusters produced are of size 13, 13 and 9. The phase allocation agrees with the network operator's allocation at all houses.

Fig 7 shows the Transformer 2 correlation matrix, reordered according to algorithm clustering.

We note that the error rate for Transformer 1 is five houses whether the aggregation level is chosen to be two, five, 10, 15, 20, 30, 60 or even 120 minutes. For example, at two and five minute aggregation, in addition, house 62 is misclassified; and at 10 to 120 minute aggregation, house 52 is misclassified.

At aggregation levels of 20 minutes and above, the Transformer 2 clustering misclassifies many houses—the error rate is 10 to 11 houses.

We also experimented with taking subsets of hours; for example, using only the hours from 3 a.m. to 6 a.m. at various aggregation levels. This did not consistently improve the error rate across both transformers.

We checked the un-ordered error rate for various input sizes in two different ways. The data for the first transformer has approximately 67 days, while the second transformer houses cover 124 days.

First, we tested the data in a sequential form—that is, for each possible day length $d \in \{1, \ldots, n\}$ where $n$ is the total number of days, we extracted all sequential rows of day length $d$ from the data, and computed the average error rate for $d$.

Second, we tested the data in random form; for each possible number of days $d$ as above, we chose ten random sequences from the data with the same number of rows as $d$ days (that is, $1440 \times d$ rows) and calculated the average error rate of the clusterings obtained using these sequences.

The results are shown in Figs 8, 9, 10 and 11 below. For transformer 1, for both sequential and random testing, the average error rate is between 4 and 5 houses. For transformer 2, for sequential testing, the error rate is zero for at least 65 days of data and for random testing the error rate for zero for at least 4 days of data. For transformer 2 with sequential testing, there is a general downward trend, with an error rate of 9.5 with one day, decreasing as more days are added.

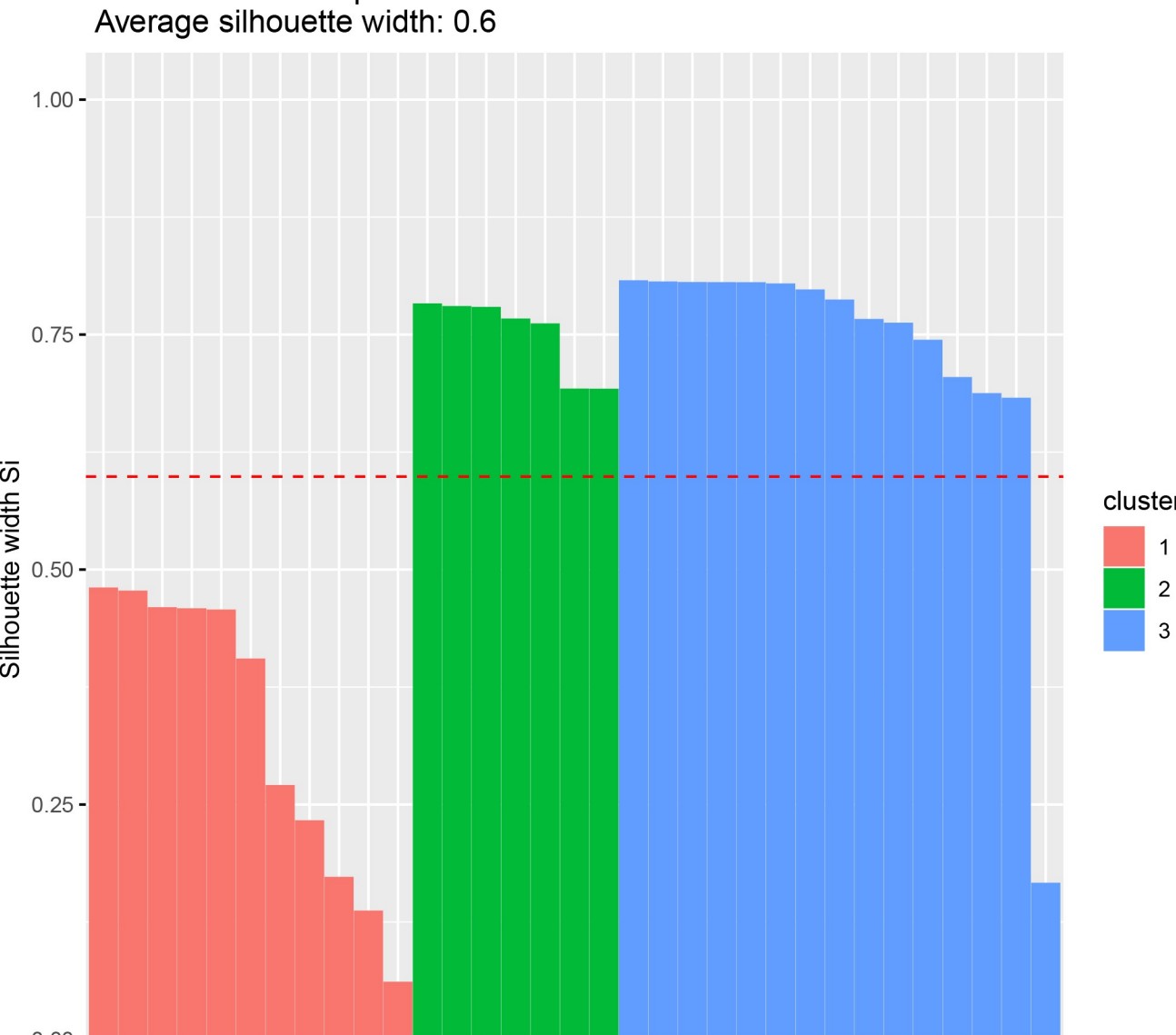

**Fig 4. Transformer 1 clusters.** Silhouette plot.

## Discussion

Remotely detecting the phase of houses on an electricity grid minimises the likelihood of loss of power to HMED users. Our phase detection algorithm correctly determined the phase of 64 of the 68 houses upon which it was tested. Of the four incorrectly determined houses, two were found to be almost certainly a record keeping error on the part of the network operator. While further testing on larger datasets is required, these early indications of performance represent over 95% accuracy. The method is promising, given that the data underpinning the algorithm (high frequency sub-metered household consumption data) will soon be ubiquitously available to network operators through smart metering or installation of network sensing devices. This

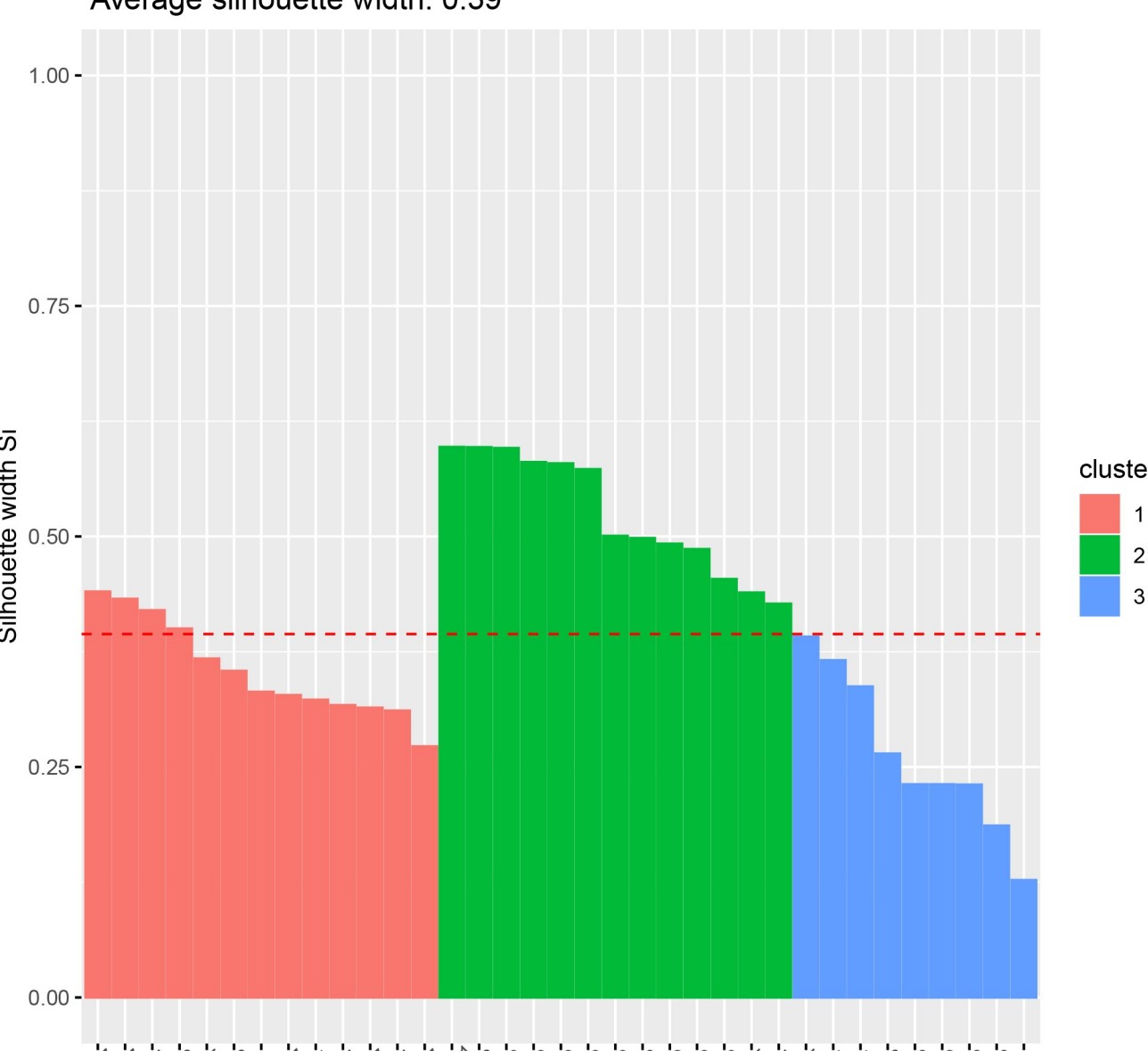

**Fig 5. Transformer 2 clusters.** Silhouette plot.

research advances prior phase detection algorithms which offer similarly high accuracy, but require transformer level energy use data [25, 26] which is not typically publicly available, and can present data interpolation issues due to the need to align timestamps on transformer and smart meter data [25].

**Benefits of algorithm and phase detection**: The ability to accurately and remotely determine the phase of a house has several direct benefits for network operators, and for users of HMEDs and other digital health technologies reliant upon uninterrupted electricity. These benefits include: improved ability to confidently forecast energy interruptions to energy vulnerable and energy dependent customers (e.g. those with HMEDs), minimise accidental

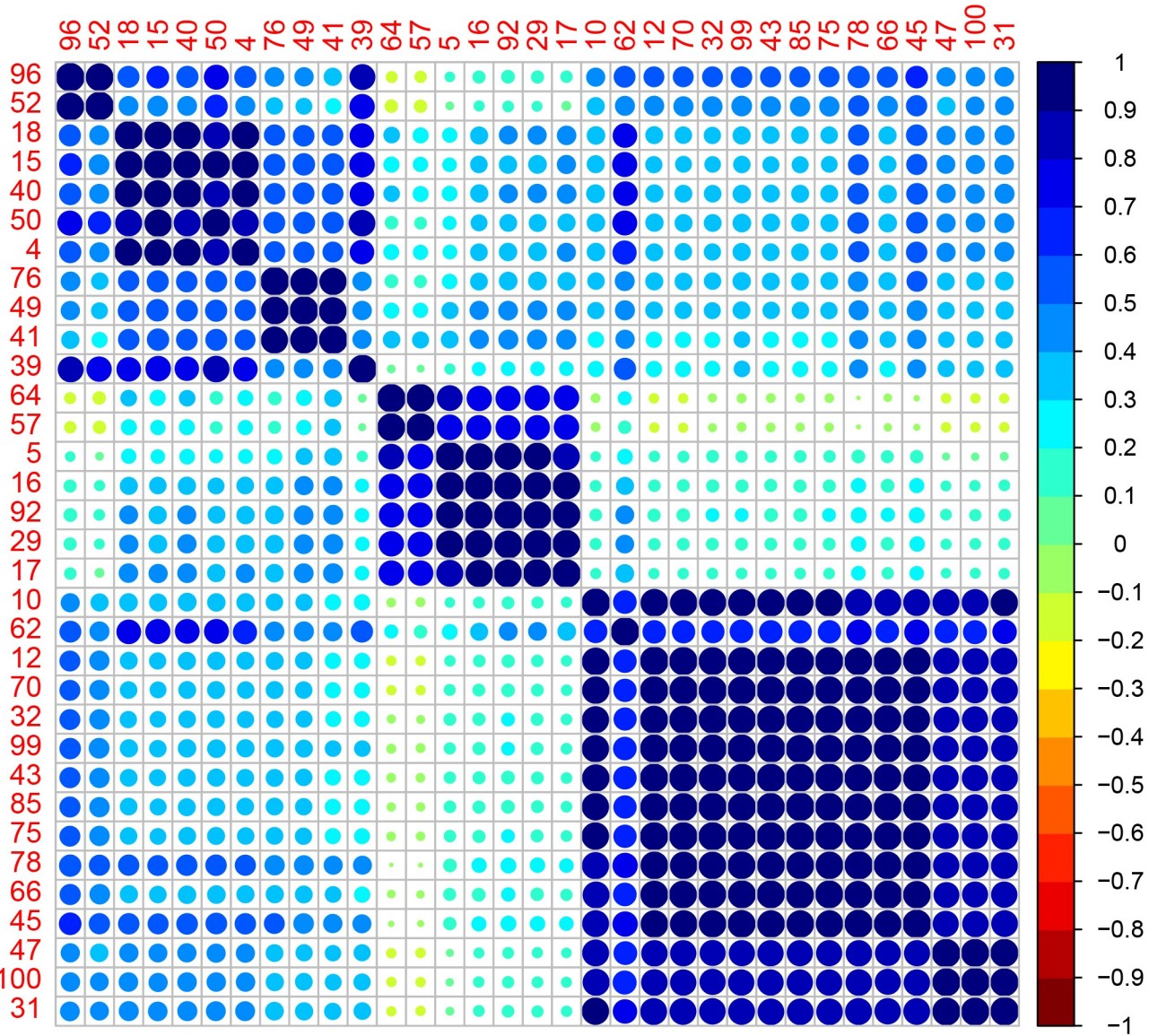

**Fig 6. Correlation plot for Transformer 1 with houses reordered according to algorithm clustering.**

disconnections and avoid fines of $20,000 (for network operators if they fail to provide at least four business days' written notice of a planned interruption to the electricity supply of a vulnerable customer [21]). An accidental disconnection resulting in loss of life from an energy-dependent customer would have profound negative economic and reputational consequences for a network operator. Additionally, having complete and accurate information of dwellings' phases within an energy grid, correlated with a register of home medical device customers, location and demographic information could assist in emergency planning by improved prediction of likely hospital admissions during a large scale network outage due to both medical device failure and heat/cold stress vulnerability among elderly demographics.

**Benefits of monitoring hardware**: Both network operators and HMED users additionally stand to benefit through the deployment of behind-the-meter technology necessary for phase detection. Behind-the-meter monitoring of energy consumption at high frequency allows for

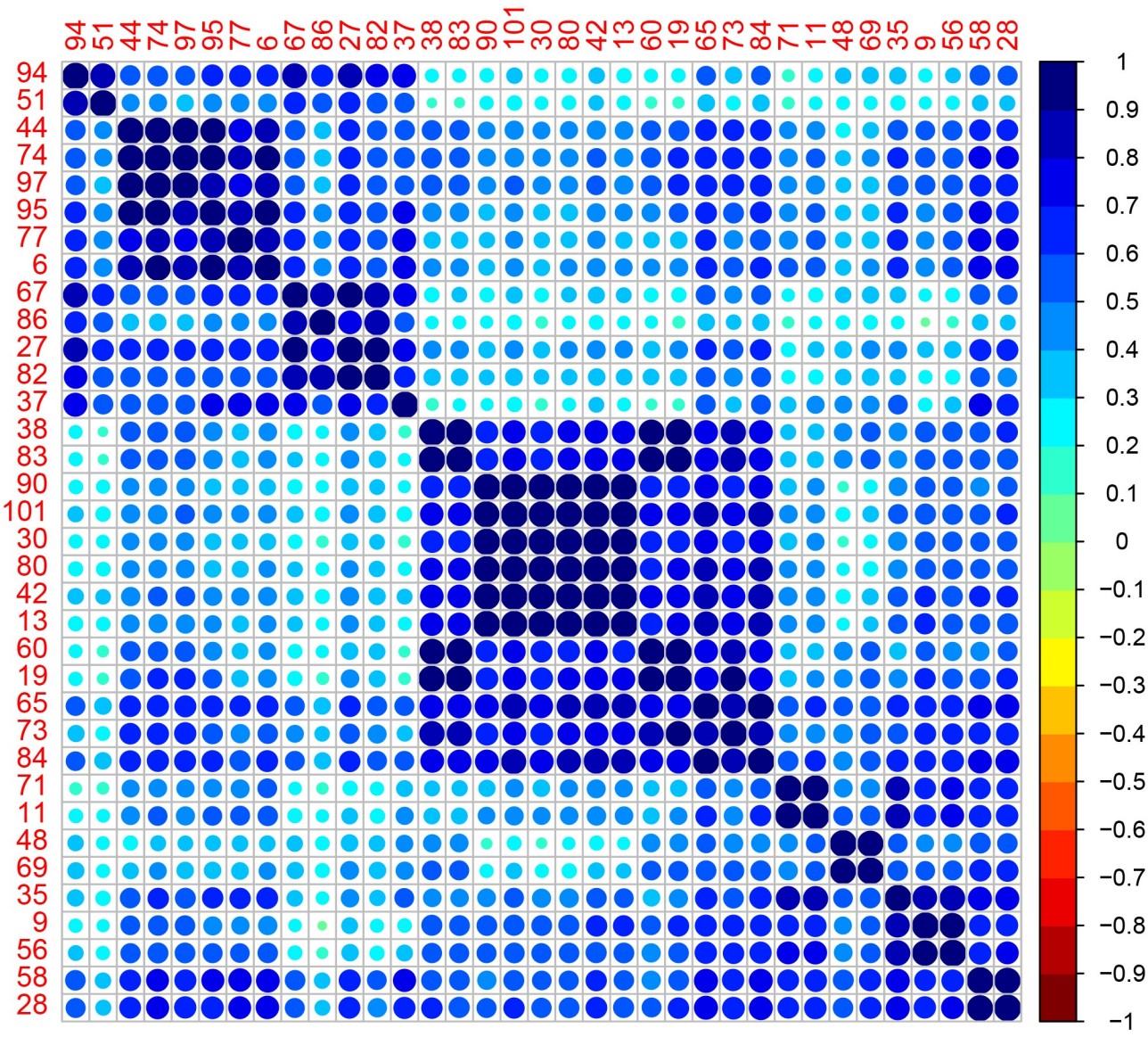

**Fig 7. Correlation plot for Transformer 2 with houses reordered according to algorithm clustering.**

the creation of energy use visualisations and improved energy use feedback information. Real-time feedback on energy use is credited with improving awareness of energy use in the home [32] and improving understanding of the relative contributors to household energy bills and reduced energy bills [33]. Electricity vulnerable and electricity dependent customers are likely to be non-standard electricity users and may benefit from a greater understanding of the factors affecting their energy consumption. Further, network operators who presently lack real time system state information beyond the bulk-substation level will benefit from the improved demand response offered by broadscale sub-metering at the household level [34].

## Collaborations towards better digital health outcomes

Advances in most digital health technologies (DHTs) are contingent upon access to stable and reliable electricity supply for powering HMED devices and charging other eHealth and

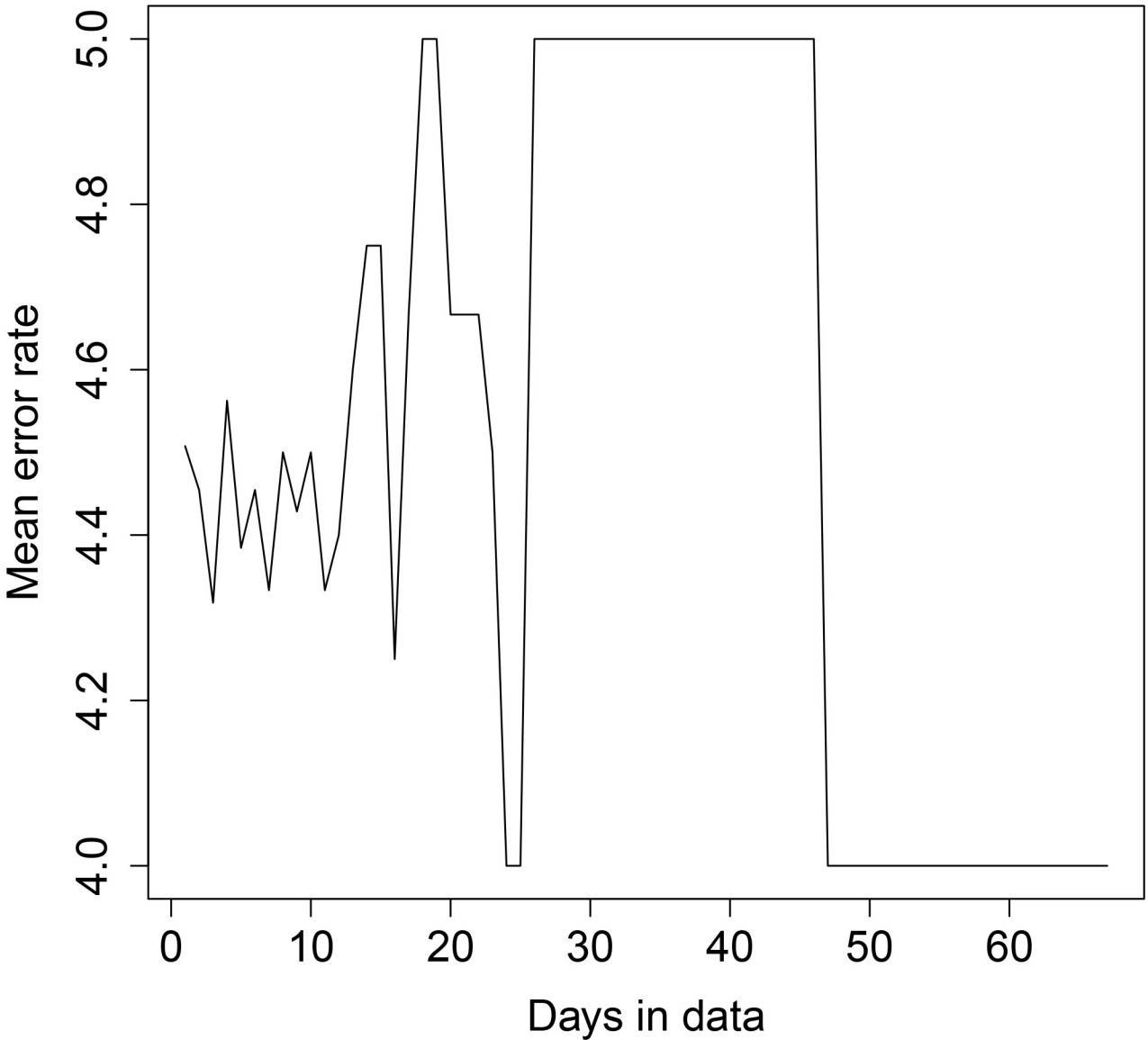

**Fig 8. Transformer 1 error rate plot—Sequential day selection.**

mHealth devices. In developed countries this access is taken for granted. It has been well documented that this access, when removed either by planned outages [18], accident or by extreme weather or natural disasters [12, 17], imposes significant stress on vulnerable individuals.

We close with some recommendations for jurisdictions without detailed records of HMED users:

1. Designers of DHTs should consider better communicating the consequences of interruptions of energy supply to users, including contingency plans, and websites/phone numbers to register their devices with network operators.

2. Network operators, DHT manufacturers, and those who distribute or hire DHTs (e.g. hospitals) should collaborate more closely in the development of a formal register of HMEDs and other digital health deployments which are electricity-critical. This register should be

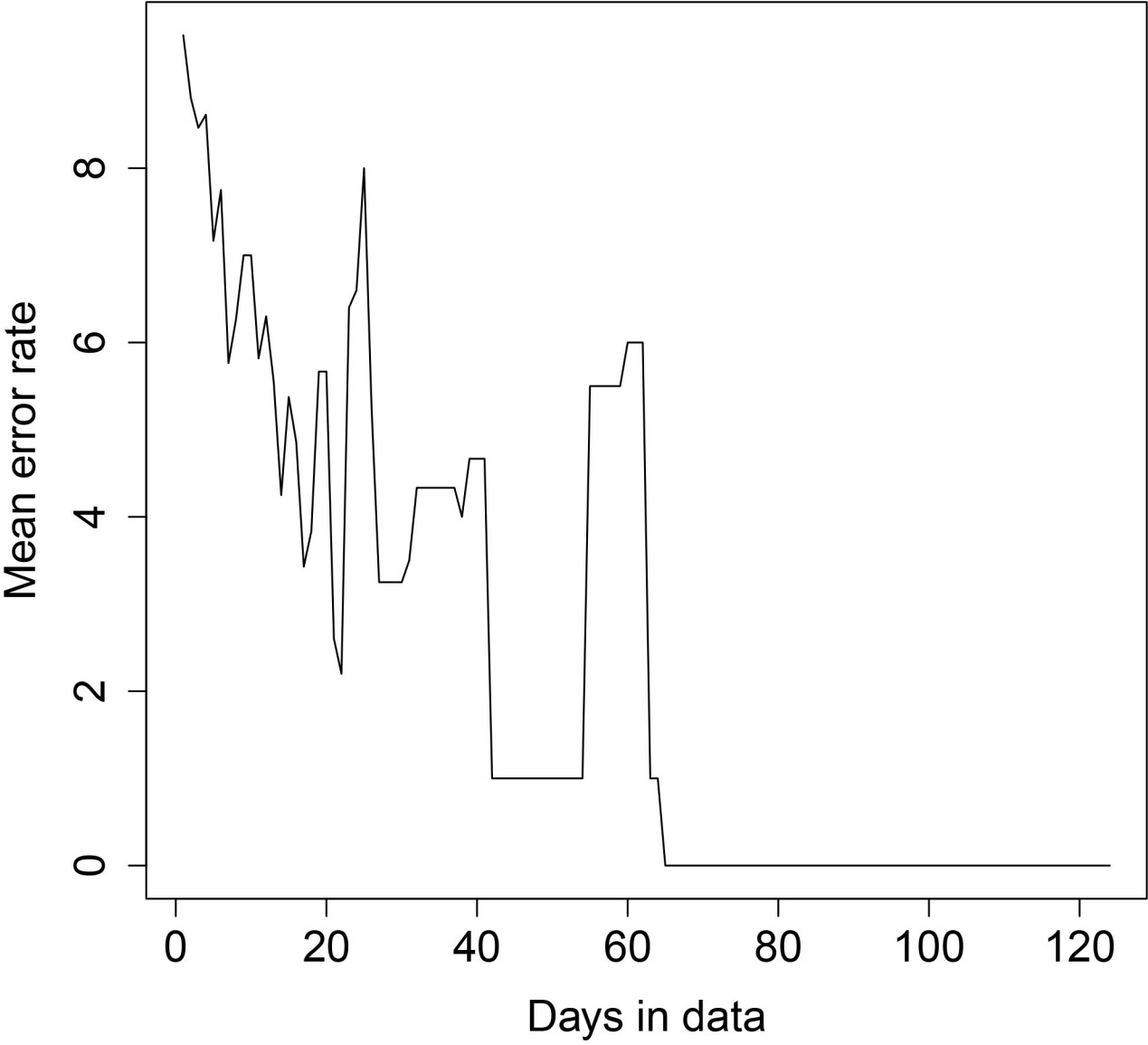

**Fig 9. Transformer 2 error rate plot—Sequential day selection.**

available and editable by energy utilities, energy networks, DHT manufacturers and other necessary stakeholders (e.g. hospitals, doctors), thereby improving information delivery to network operators about the number, type and location of HMEDs that are deployed. This information could be fed into phase detection algorithms to further lower the likelihood of accidental disconnection. Consideration of personal data management and confidentiality are necessary here; however, moving the onus of registration with the network operator or energy utility from user to those who distribute or deploy HMEDs (e.g. hospitals, doctors etc) may increase the accuracy of location-based information on HMED deployment.

3. Further work is required to increase the accuracy of phase detection, to improve certainty related to power interruptions. In order to improve the accuracy of the clustering algorithm,

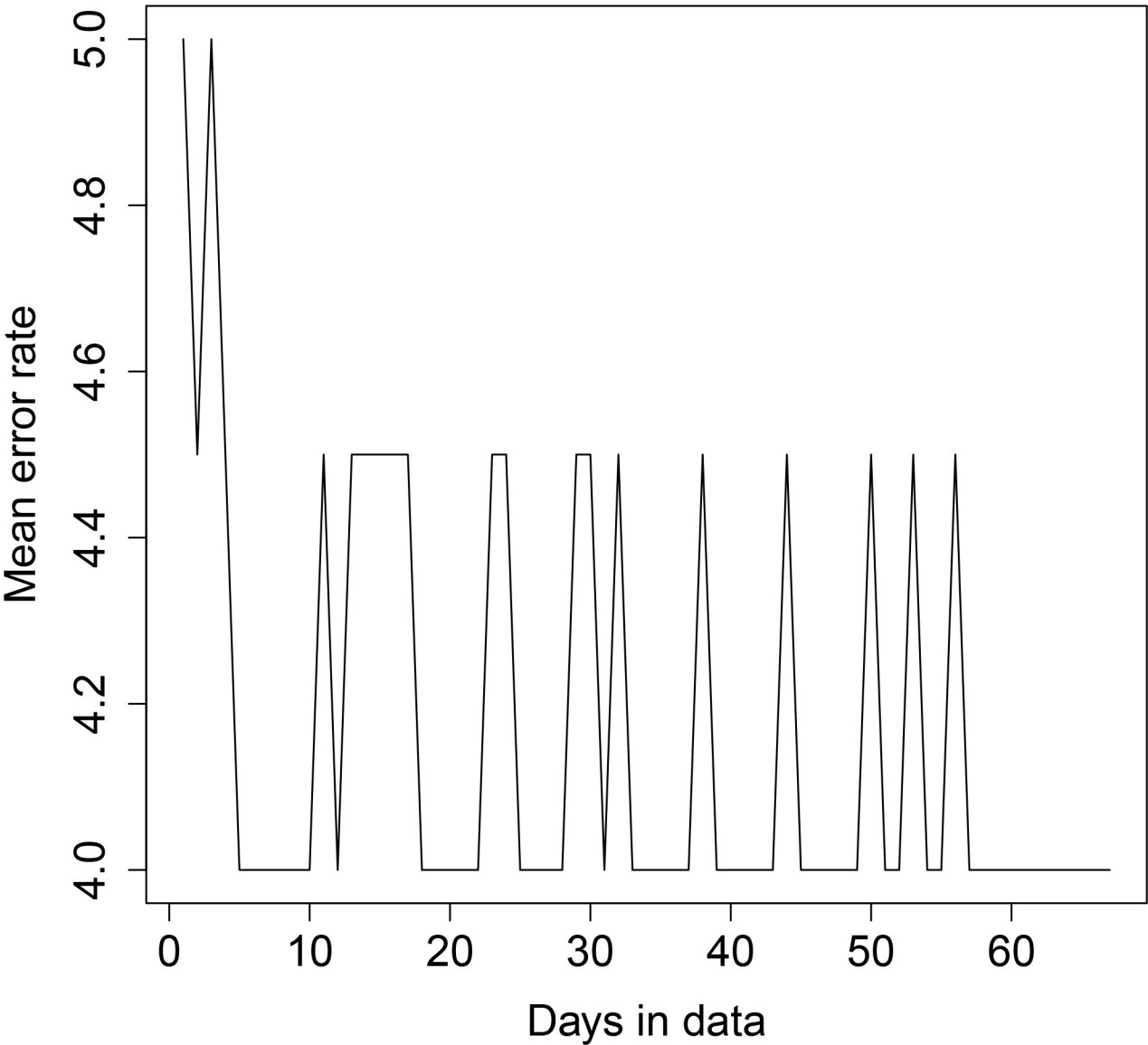

**Fig 10. Transformer 1 error rate plot—Random day selection.**

more phase identification data should be obtained, and a procedure developed to output a "confidence threshold" for correct phase identification, while using other information such as distance from the transformer and voltage at each house, together with testing different temporal aggregation levels and time slices of the day. When this threshold has been developed, in order to obtain a perfect phase identification, it will be necessary to test only a few houses attached to the transformer, such as representatives from each cluster, and houses that fall below this threshold. Since the data of this paper was collected, the resolution of the voltage data collected by the Power Analyzer device has improved from 1 V to 0.1 V. As Wang et al [22] noted, more granular voltage time series led to higher phase identification accuracy and we would also expect that this more accurate voltage data would improve accuracy.

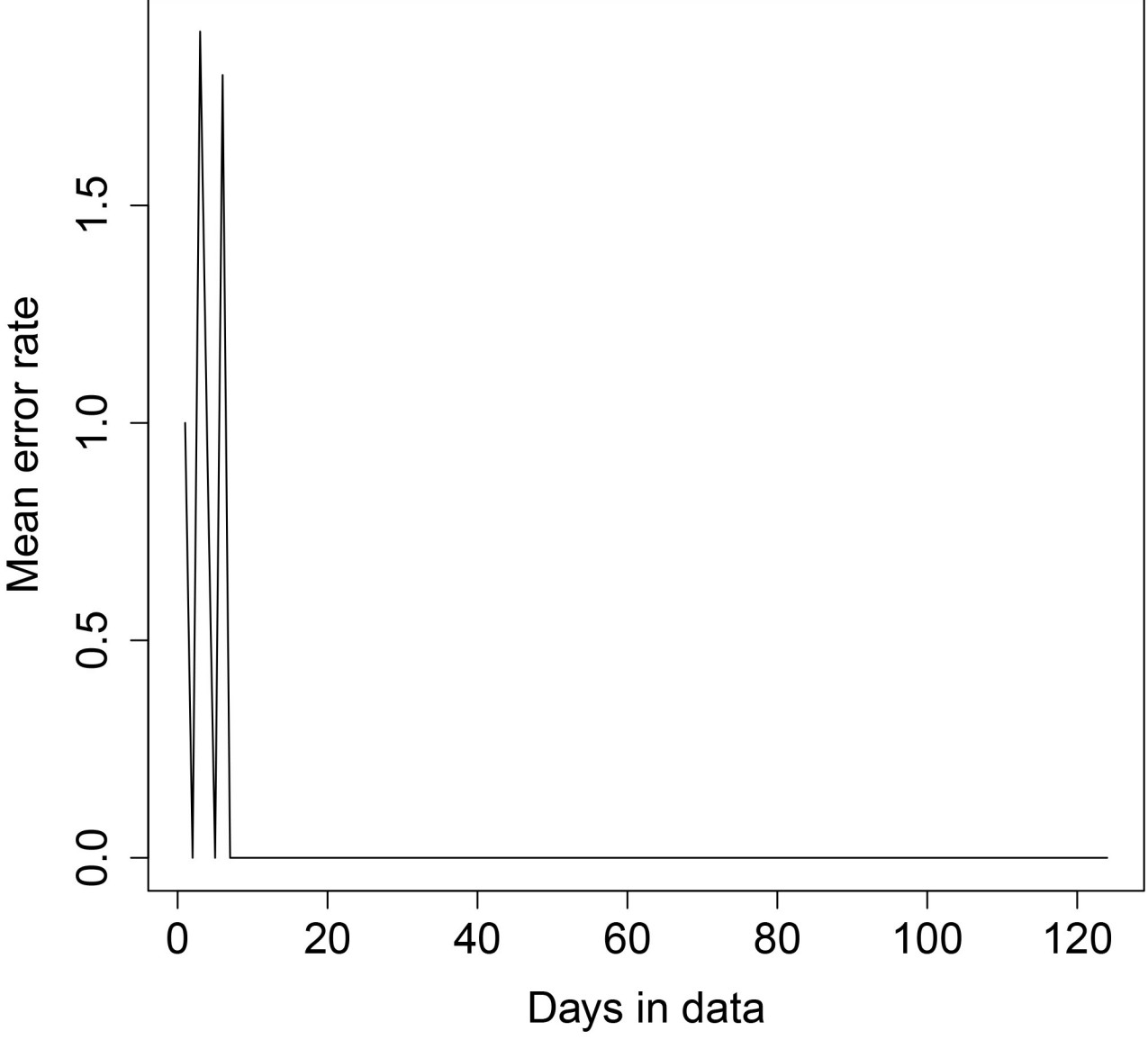

**Fig 11. Transformer 2 error rate plot—Random day selection.**

In concluding, we argue that because continuation of advances in DHT are contingent upon reliable electricity supply, future work in digital health should consider power use needs beyond individual devices, and seek collaborations with researchers engaged in energy network management and emergency preparedness. To these ends, this paper has provided a proof of concept for a phase detection with tangible benefits to energy network operators, at home medical device users, digital health researchers and those engaged in emergency preparedness.

## Supporting information

**S1 Data.**
(DOCX)

## Author Contributions

**Data curation:** Richard Bean.

**Investigation:** Richard Bean, Stephen Snow.

**Software:** Richard Bean.

**Writing – original draft:** Richard Bean, Stephen Snow, Mashhuda Glencross.

**Writing – review & editing:** Stephen Viller, Neil Horrocks.

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
