## [Decision Letter · Decision Letter 0]

25 Feb 2020

PONE-D-20-02498

Keeping the Power on to Home Medical Devices

PLOS ONE

Dear Dr Bean,

Thank you for submitting your manuscript to PLOS ONE. After careful consideration, we feel that it has merit but does not fully meet PLOS ONE’s publication criteria as it currently stands. Therefore, we invite you to submit a revised version of the manuscript that addresses the points raised during the review process.

We would appreciate receiving your revised manuscript by Apr 10 2020 11:59PM. To enhance the reproducibility of your results, we recommend that if applicable you deposit your laboratory protocols in protocols.io, where a protocol can be assigned its own identifier (DOI) such that it can be cited independently in the future. For instructions see: http://journals.plos.org/plosone/s/submission-guidelines#loc-laboratory-protocols

We look forward to receiving your revised manuscript.

Kind regards,

Talib Al-Ameri, Ph.D

Academic Editor

PLOS ONE

Journal Requirements:

"No response"

Please provide an amended Funding Statement that declares *all* the funding or sources of support received during this specific study (whether external or internal to your organization) as detailed online in our guide for authors at http://journals.plos.org/plosone/s/submit-now.  Please state what role the funders took in the study.  If any authors received a salary from any of your funders, please state which authors and which funder. If the funders had no role, please state: "The funders had no role in study design, data collection and analysis, decision to publish, or preparation of the manuscript."

"No response"

5. Please ensure that you refer to Figures 3, 5, 6, 7, 8, 9 in your text as, if accepted, production will need this reference to link the reader to the figure.

Reviewers' comments:

Reviewer's Responses to Questions

**Comments to the Author**

1. Is the manuscript technically sound, and do the data support the conclusions?

Reviewer #1: No

Reviewer #2: Yes

Reviewer #3: Partly

2. Has the statistical analysis been performed appropriately and rigorously? 

Reviewer #1: N/A

Reviewer #2: I Don't Know

Reviewer #3: I Don't Know

3. Have the authors made all data underlying the findings in their manuscript fully available?

Reviewer #1: No

Reviewer #2: Yes

Reviewer #3: No

4. Is the manuscript presented in an intelligible fashion and written in standard English?

Reviewer #1: No

Reviewer #2: Yes

Reviewer #3: Yes

5. Review Comments to the Author

Reviewer #1: The novelty of this paper if any have not been significant. Substantial scientific aspects are missing in the paper such as the design of the proposed algorithm and hardware of the system. My recommendation is the paper is not suitable to publish in PLOS ONE (rejected).

Reviewer #2: This article addressed the power supply issue regarding home medical devices which has practical values. The work can be of certain reference in the area. I would like to recommend consider it for acceptance after further improvement.

Here are some of my impression about the work:

1. The present study appears some what too engineering which needs fruther in-depth improvement especially on fundamental parts.

2. Some earlier literatures on mobile health care are needed.

3. Originality of the curent work is not very clear. Power is a necessity and tremendous devices are available for such needs.

4. Adding some more basic innovative theoretical strategies will enhance curent work.

5. Unique feature of current energy supply perhaps can still be distinguished with existing one.

Reviewer #3: The authors present what appears to be a very novel idea with useful application. However, the development and operation of the phase algorithm, at least as it's written here is seems vague. In general, a more thorough presentation of the algorithm and analysis in reference to the actual operator measurements are needed. Please include some calculations and/or code on how, for example the error rates and plots were obtained. Figures 2 and 4 should be clearer/labeled better.

It would be most convincing to provide a map of the houses sampled, where the algorithm's phase is displayed, as well as the actual phase based on operator records. Authors should provide a copy of the operator records, perhaps as supplemental data.

6. PLOS authors have the option to publish the peer review history of their article (what does this mean?). If published, this will include your full peer review and any attached files.

Reviewer #1: No

Reviewer #2: No

Reviewer #3: No

---

## [Author Response · Author response to Decision Letter 0]

23 Apr 2020

The response has been provided in the attached DOC file.

---

## [Decision Letter · Decision Letter 1]

25 May 2020

PONE-D-20-02498R1

Keeping the power on to home medical devices

PLOS ONE

Dear Dr. Bean,

Thank you for submitting your manuscript to PLOS ONE. After careful consideration, we feel that it has merit but does not fully meet PLOS ONE’s publication criteria as it currently stands. Therefore, we invite you to submit a revised version of the manuscript that addresses the points raised during the review process.

We look forward to receiving your revised manuscript.

Kind regards,

Talib Al-Ameri, Ph.D

Academic Editor

PLOS ONE

Reviewers' comments:

Reviewer's Responses to Questions

**Comments to the Author**

1. If the authors have adequately addressed your comments raised in a previous round of review and you feel that this manuscript is now acceptable for publication, you may indicate that here to bypass the “Comments to the Author” section, enter your conflict of interest statement in the “Confidential to Editor” section, and submit your "Accept" recommendation.

Reviewer #1: (No Response)

Reviewer #2: All comments have been addressed

Reviewer #3: All comments have been addressed

2. Is the manuscript technically sound, and do the data support the conclusions?

Reviewer #1: No

Reviewer #2: Partly

Reviewer #3: (No Response)

3. Has the statistical analysis been performed appropriately and rigorously? 

Reviewer #1: No

Reviewer #2: N/A

Reviewer #3: (No Response)

4. Have the authors made all data underlying the findings in their manuscript fully available?

Reviewer #1: No

Reviewer #2: Yes

Reviewer #3: (No Response)

5. Is the manuscript presented in an intelligible fashion and written in standard English?

Reviewer #1: No

Reviewer #2: Yes

Reviewer #3: (No Response)

6. Review Comments to the Author

Reviewer #1: (No Response)

Reviewer #2: The revised article has partially addressed the comments. A main concern is that originality can still be strengthened to fullfil the quanlity standard of PLoS ONE as could as possible. Anyway, I recommend accept the article for publication.

Reviewer #3: (No Response)

7. PLOS authors have the option to publish the peer review history of their article (what does this mean?). If published, this will include your full peer review and any attached files.

Reviewer #1: No

Reviewer #2: No

Reviewer #3: No

---

## [Author Response · Author response to Decision Letter 1]

31 May 2020

Response to Reviewer #1 concerns in new "Response to Reviewers" letter - there is no new detail in reviewer comments, so we cannot do anything. We have asked the editor for a suggestion on what to do.

From response:

Reviewer #1 is actually incorrectly stating that details about the hardware (and in fact the algorithm) are missing as we have stated which commercial hardware we are using and we have disclosed how the algorithm works (on pages 6-8) just not a full disclosure and this is because there is a patent pending – there are a number of pages of detail. What would the editor recommend we do to address a factually incorrect comment from reviewer #1?

---

## [Decision Letter · Decision Letter 2]

9 Jun 2020

Keeping the power on to home medical devices

PONE-D-20-02498R2

Dear Dr. Bean,

We’re pleased to inform you that your manuscript has been judged scientifically suitable for publication and will be formally accepted for publication once it meets all outstanding technical requirements.

Kind regards,

Talib Al-Ameri, Ph.D

Academic Editor

PLOS ONE

Additional Editor Comments (optional):

Reviewers' comments:

Reviewer's Responses to Questions

**Comments to the Author**

1. If the authors have adequately addressed your comments raised in a previous round of review and you feel that this manuscript is now acceptable for publication, you may indicate that here to bypass the “Comments to the Author” section, enter your conflict of interest statement in the “Confidential to Editor” section, and submit your "Accept" recommendation.

Reviewer #1: All comments have been addressed

2. Is the manuscript technically sound, and do the data support the conclusions?

Reviewer #1: Yes

3. Has the statistical analysis been performed appropriately and rigorously? 

Reviewer #1: Yes

4. Have the authors made all data underlying the findings in their manuscript fully available?

Reviewer #1: Yes

5. Is the manuscript presented in an intelligible fashion and written in standard English?

Reviewer #1: Yes

6. Review Comments to the Author

Reviewer #1: According to the intellectual property where the authors could not present more details about their work. I have no further comments and I recommend accepting the paper.

7. PLOS authors have the option to publish the peer review history of their article (what does this mean?). If published, this will include your full peer review and any attached files.

Reviewer #1: Yes: Sadik Kamel Gharghan

---

## [Editor Report · Acceptance letter]

18 Jun 2020

PONE-D-20-02498R2 

Keeping the power on to home medical devices 

Dear Dr. Bean:

I'm pleased to inform you that your manuscript has been deemed suitable for publication in PLOS ONE. Congratulations! Your manuscript is now with our production department. 

Kind regards, 

on behalf of

Dr. Talib Al-Ameri 

Academic Editor

PLOS ONE